# Whole-Genome Sequencing for Resistance Prediction and Transmission Analysis of *Mycobacterium tuberculosis* Complex Strains from Namibia

M. Claassens,[a] V. Dreyer,[b,e] E. Nepolo,[a] Q. Mokomele,[c] G. van Rooyen,[c] N. Ruswa,[d] G. Günther,[a,f,g] S. Niemann[a,b,e]

[a]Department of Human, Biological and Translational Medical Sciences, School of Medicine, University of Namibia, Windhoek, Namibia
[b]Molecular and Experimental Mycobacteriology, Research Center Borstel, Borstel, Germany
[c]Namibia Institute of Pathology Limited, Windhoek, Namibia
[d]National Tuberculosis and Leprosy Programme, Windhoek, Namibia
[e]German Center for Infection Research, Partner Site Hamburg-Lübeck-Borstel-Riems, Borstel, Germany
[f]Department of Pulmonology and Allergology, Inselspital Bern, Bern University Hospital, University of Bern, Bern, Switzerland
[g]Katutura State Hospital, Ministry of Health and Social Services, Windhoek, Namibia

M. Claassens, V. Dreyer, and E. Nepolo contributed equally to this article. Author order was determined alphabetically.
G. Günther and S. Niemann contributed equally.

**ABSTRACT** Namibia is among 30 countries with a high burden of tuberculosis (TB), with an estimated incidence of 460 per 100,000 population and around 800 new multidrug-resistant (MDR) TB cases per year. Still, data on the transmission and evolution of drug-resistant *Mycobacterium tuberculosis* complex (Mtbc) strains are not available. Whole-genome sequencing data of 136 rifampicin-resistant (RIFr) Mtbc strains obtained from 2016 to 2018 were used for phylogenetic classification, resistance prediction, and cluster analysis and linked with phenotypic drug susceptibility testing (pDST) data. Roughly 50% of the strains investigated were resistant to all first-line drugs. Furthermore, 13% of the MDR Mtbc strains were already pre-extensively drug resistant (pre-XDR). The cluster rates were high, at 74.6% among MDR and 85% among pre-XDR strains. A significant proportion of strains had borderline resistance-conferring mutations, e.g., *inhA* promoter mutations or *rpoB* L430P. Accordingly, 25% of the RIFr strains tested susceptible by pDST. Finally, we determined a potentially new bedaquiline resistance mutation (Rv0678 D88G) occurring in two independent clusters. High rates of resistance to first-line drugs in line with emerging pre-XDR and likely bedaquiline resistance linked with the ongoing recent transmission of MDR Mtbc clones underline the urgent need for the implementation of interventions that allow rapid diagnostics to break MDR TB transmission chains in the country. A borderline RIFr mutation in the dominant outbreak strain causing discrepancies between phenotypic and genotypic resistance testing results may require breakpoint adjustments but also may allow individualized regimens with high-dose treatment.

**IMPORTANCE** The transmission of drug-resistant tuberculosis (TB) is a major problem for global TB control. Using genome sequencing, we showed that 13% of the multidrug-resistant (MDR) *M. tuberculosis* complex strains from Namibia are already pre-extensively drug resistant (pre-XDR), which is substantial in an African setting. Our data also indicate that the ongoing transmission of MDR and pre-XDR strains contributes significantly to the problem. In contrast to other settings with higher rates of drug resistance, we found a high proportion of strains having so-called borderline low-level resistance mutations, e.g., *inhA* promoter mutations or *rpoB* L430P. This led to the misclassification of 25% of the rifampicin-resistant strains as susceptible by phenotypic drug susceptibility testing. This observation potentially allows individualized regimens with high-dose treatment as a potential option for patients with few treatment options. We also found a potentially new bedaquiline resistance mutation in *rv0678*.

Address correspondence to S. Niemann, sniemann@fz-borstel.de.

The authors declare no conflict of interest.

**KEYWORDS** Namibia, drug resistance, low-level resistance, whole-genome sequencing

Drug-resistant (DR), multidrug-resistant (MDR) (resistance to at least isoniazid [INH] and rifampicin [RIF]), pre-extensively drug resistant (pre-XDR) (MDR strains additionally resistant to at least one fluoroquinolone [FQ]), and XDR (additional resistance to one World Health Organization [WHO] group A drug) *Mycobacterium tuberculosis* complex (Mtbc) strains represent a serious challenge for local and global tuberculosis (TB) control (1). Poor case detection and management, pharmacokinetic variability, limited treatment options, and ongoing transmission have fostered the MDR/pre-XDR/XDR TB epidemic, leading to an estimated total global number of more than half a million new RIF-resistant (RIFr) TB cases, of which more than 75% are MDR (2, 3). The MDR TB prevalence has increased to more than 30% among new cases in several high-burden countries, and XDR TB cases (old WHO classification) have been reported from more than 120 countries (2, 3).

Namibia is a WHO high-burden country for TB and TB-HIV. In 2020, the TB incidence was estimated to be 460/100,000, equating to ~12,000 new cases, annually (4). A TB prevalence survey in 2018 documented 465/100,000 prevalent cases (5). A recent drug resistance survey revealed MDR TB rates of 4.5% among new patients and 7.9% among previously treated patients (6). The overall number of new MDR TB cases per year is estimated to be around 800, while only 213 cases were detected in 2020 (4). Nearly half (46.6%) of the patients with MDR TB were coinfected with HIV (6). In a 2018 MDR TB cohort, 64% of the MDR TB patients had a successful treatment outcome (4).

Despite the high TB and MDR TB case numbers and poor outcomes for MDR TB patients likely fostering the spread of MDR TB, the factors driving the MDR TB epidemic in Namibia remain poorly defined, and data on the transmission dynamics of MDR Mtbc strains are not available. However, precise data on the factors driving the MDR/XDR TB epidemic in the country, including data on the molecular drug resistance determinants as well the transmission dynamics and evolution of MDR Mtbc strains, are crucial for developing more effective TB control strategies.

To tackle these questions, we performed whole-genome sequencing (WGS) of a convenient sample of RIFr Mtbc strains obtained from the Namibia Institute of Pathology Limited (NIP) from 2016 to 2018. WGS data of the Mtbc isolates have been used to perform phylogenetic classification, resistance prediction, and cluster analysis. We also screened for dominant strains and/or mutation signatures associated with DR TB and for mutations known to increase the fitness and, presumably, the transmission potential of MDR TB strains.

## RESULTS

**Study population.** WGS was successfully performed for 136 RIFr Mtbc isolates obtained in 2016, 2017, and 2018. WGS data were used to perform basic phylogenetic classification, resistance prediction, and cluster analysis (see Table S2 in the supplemental material). Basic characteristics are shown in Table 1. Data are summarized in Table S2. The mean age of participants was 37.7 years (standard deviation [SD], 12.1 years), and the majority of the participants ($n = 81$; 59.6%) were male. The majority ($n = 113$; 83.1%) were Namibian, while the other participants came from Angola. Overall, half of the study population ($n = 72$; 52.9%) were new TB cases, while 40.2% ($n = 55$) were HIV positive, of whom 48 (87.3%) were on antiretroviral therapy.

**Population structure and drug resistance mutations.** In total, 3,402 informative single nucleotide polymorphisms (SNPs) differentiating any of the 136 Mtbc isolates were identified and used to calculate a maximum likelihood (ML) tree (Fig. 1). WGS data were used to classify the strains into Mtbc lineages using canonical SNPs described previously (7–9).

Virtually all Mtbc strains were classified as belonging to the Euro-American lineage (L4; $n = 132$); just 4 strains belonged to the Beijing lineage (L2). The majority of L4 strains could be further classified as belonging to LAM (4.3; $n = 102$), followed by the

**TABLE 1** Descriptive characteristics of DR TB patients included in the pilot WGS study ($n = 136$)[a]

| Variable | No. of patients | Proportion of patients (%) |
|---|---|---|
| Age (yrs) | | |
| 14–24 | 17 | 13 |
| 25–34 | 41 | 30 |
| 35–44 | 42 | 31 |
| 45–54 | 24 | 18 |
| >55 | 11 | 8 |
| Missing | 1 | 1 |
| | | |
| Sex | | |
| Female | 55 | 40 |
| Male | 81 | 60 |
| | | |
| Nationality | | |
| Namibian | 113 | 83 |
| Angolan | 23 | 17 |
| | | |
| Type of treatment | | |
| New | 72 | 53 |
| Retreatment after default | 10 | 7 |
| Retreatment after failure | 13 | 10 |
| Retreatment after reoccurrence | 33 | 24 |
| Missing | 8 | 6 |
| | | |
| WGS classification | | |
| MDR | 109 | 80 |
| RIFr | 14 | 10 |
| Pre-XDR | 13 | 10 |
| | | |
| WGS lineage | | |
| Beijing | 4 | 3 |
| Euro-American | 1 | 1 |
| H37Rv-like | 1 | 1 |
| LAM | 102 | 75 |
| S type | 5 | 4 |
| X type | 13 | 10 |
| Mainly T | 10 | 7 |
| | | |
| HIV status | | |
| Positive | 55 | 40 |
| Negative | 72 | 53 |
| Missing | 9 | 7 |
| | | |
| On ART ($n = 55$) | | |
| Yes | 48 | 87 |
| No | 5 | 9 |
| Missing | 2 | 4 |
| | | |
| Region | | |
| Erongo | 6 | 4 |
| Hardap | 5 | 4 |
| Karas | 3 | 2 |
| Kavango | 13 | 10 |
| Khomas | 26 | 19 |
| Kunene | 1 | 1 |
| Ohangwena | 21 | 15 |
| Omaheke | 2 | 1 |
| Omusati | 8 | 6 |
| Oshana | 15 | 11 |
| Oshikoto | 5 | 4 |
| Otjozondjupa | 13 | 10 |
| Zambezi | 8 | 6 |
| Missing | 10 | 7 |

[a]$n = 136$ for all variables apart from "on ART" (antiretroviral therapy), which is for the HIV-positive group only ($n = 55$).

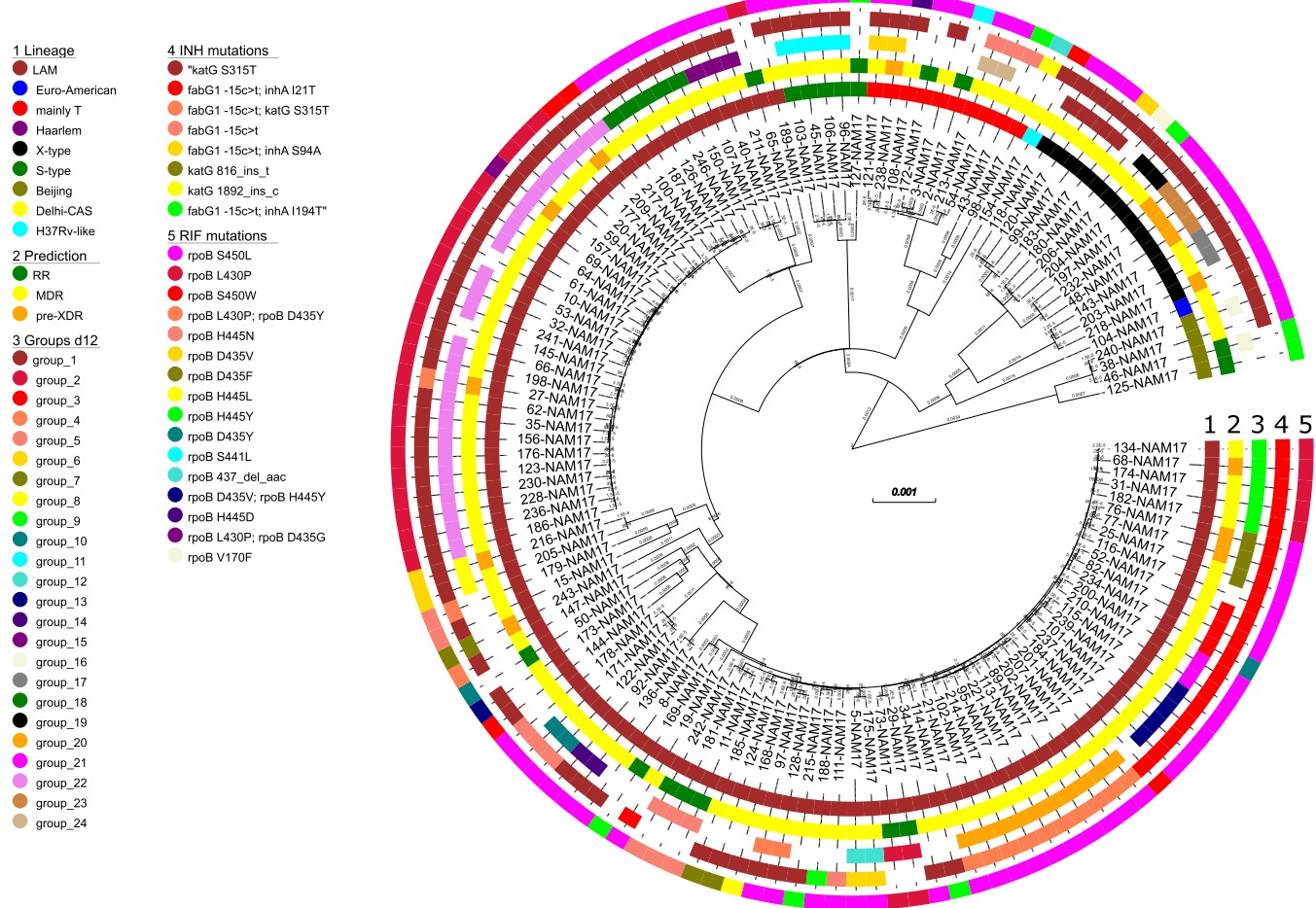

**FIG 1** Maximum likelihood phylogeny based on the concatenated SNP sequence from 136 Mtbc strains. The concatenated SNP sequence consists of 3,402 parsimony-informative sites, 2,029 singleton sites, and 0 constant sites; mutations related to the respective drugs and resistance status are color-coded and expressed as annotation rings on the tree. RR, rifampicin resistant; MDR, multidrug resistant; pre-XDR, pre-extensively drug resistant; INH, isoniazid; RIF, rifampicin; d12, distance of ≤12 SNPs.

Mtbc lineages X type (4.1; $n = 13$), mainly T (4.8; $n = 10$), and S type (4.4; $n = 5$). The SNP-based lineage classification is in full concordance with the ML tree-based phylogeny (Fig. 1). A high-resolution lineage classification of the Mtbc strains revealed the following distribution: Beijing 2.2.1 ($n = 4$), Euro-American 4.1.2 ($n = 1$), H37Rv-like 4.9 ($n = 1$), LAM 4.3.2 ($n = 33$), LAM 4.3.2.1 ($n = 4$), LAM 4.3.3 ($n = 4$), LAM 4.3.4 ($n = 4$), LAM 4.3.4.1 ($n = 53$), LAM 4.3.4.2 ($n = 3$), LAM 4.3.4.2.1 ($n = 1$), mainly T 4.8 ($n = 10$), S type 4.4.1.1 ($n = 5$), X type 4.1.1.1 ($n = 10$), and X type 4.1.1.3 ($n = 3$) (Table S2).

We then performed genotypic resistance prediction based on high-confidence resistance mutations (Tables S1 and S2). Other mutations in resistance genes are listed in Table S2 but were not considered resistance conferring at first glance. Overall, 122 of the 136 RIFr strains (89.7%) were resistant to INH and classified as at least MDR. Furthermore, 88 (64.5%) Mtbc strains were resistant to ethambutol (EMB), and 83 (61.0%) were resistant to pyrazinamide (PZA). Thirteen of the MDR strains (13/122; 9.6%) were further classified as pre-XDR based on an additional resistance to one fluoroquinolone (FQ; $n = 12$) or bedaquiline (BDQ)/clofazimine ($n = 1$) (Fig. 1; Table S2). This strain showed a deletion in *rv0678* likely leading to resistance to clofazimine and bedaquiline. In total, 66 strains (48.5%) were resistant to all first-line drugs, of which 12 (8.8% of the total) were pre-XDR and 6 (4.4% of the total) had additional resistance to an injectable drug based on mutations in *rrs* or the *eis* promoter region (Table S2). No mutations related to resistance to linezolid were detected.

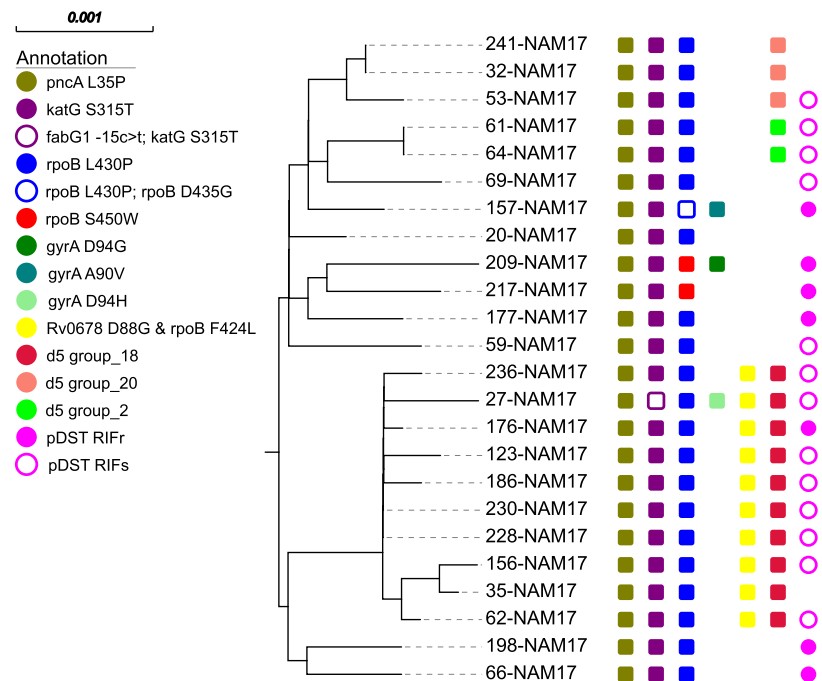

**FIG 2** Maximum likelihood phylogeny of SNP group 22 based on the concatenated SNP sequence from 24 Mtbc strains of group 22. The concatenated SNP sequence consists of 21 parsimony-informative sites, 69 singleton sites, and 0 constant sites; mutations related to the respective drugs and resistance status are color-coded and expressed as annotation columns on the tree. pDST, phenotypic drug susceptibility testing; d5, distance of ≤5 SNPs; RIFs, rifampicin sensitive.

The majority of RIF resistance-mediating mutations were found at *rpoB* codon position 450 (*n* = 74; 54.4%), mainly *rpoB* S450L. Besides, we found high *rpoB* mutation diversity. In total, 35 (25.7%) Mtbc strains had the mutation *rpoB* L430P, which has been described as a borderline RIFr mutation difficult to detect by phenotypic drug susceptibility testing (pDST) (10). Accordingly, 20 of the 26 Mtbc strains, for which pDST data were available and *rpoB* L430P was the only mutation in *rpoB*, tested susceptible (Table S2). Two strains that, besides *rpoB* L430P, had the additional canonical RIFr mutation *rpoB* D435G or *rpoB* D435Y, respectively, tested RIF resistant (Table S2).

All strains with *rpoB* L430P belonged to the LAM lineage; however, they were grouped into different sublineages, 4.3.2 (*n* = 25), 4.3.4.1 (*n* = 8), 4.3.4.2 (*n* = 1), and 4.3.2.1 (*n* = 1), and also belonged to different genome clusters (Fig. 1). This demonstrates that *rpoB* L430P is homoplastic and points toward the positive selection of this mutation in the population investigated. Besides *rpoB* L430P, further borderline RIFr mutations such as *rpoB* H445D/L/N/Y and *rpoB* D435Y were found in 17 other Mtbc strains, of which one-third (5 out of 15 with pDST data available) were susceptible by pDST (Table S2) (10). In total, one-quarter (29 out of 115) of the Mtbc strains with pDST data available tested susceptible, of which 25 had a borderline resistance-conferring mutation (Table S2).

Considering INH resistance, 89 strains had the *katG* S315T mutation (65.4% of all strains and 81.6% of MDR cases), of which 12 had an additional *fabG1* c-15t mutation, also described as a borderline resistance mutation (Table S2) (11). Furthermore, 31 Mtbc strains had a *fabG1* c-15t mutation without *katG* S315T, of which 24 had a combination with an additional mutation in *inhA*, with *inhA* I21T being the most prevalent mutation, occurring in 21 strains (Fig. 1; Table S2). However, different from strains with low-level *rpoB* mutations, pDST for INH worked well, as strains with the *fabG1* c-15t mutation mainly tested resistant (Table S2).

**Genome-based cluster analysis.** Based on a maximum distance of 12 SNPs (d12), 98 out of 136 RIFr (72.0%) and 91 out of 122 MDR (74.5%) Mtbc strains were grouped into 24 clusters ranging in size from 2 to 25 isolates (Fig. 2; Table S2). Eleven out of the 13 pre-XDR (84.6%) strains were grouped into six clusters (Fig. 1; Table S2).

**Map of admin regions inside Namibia**

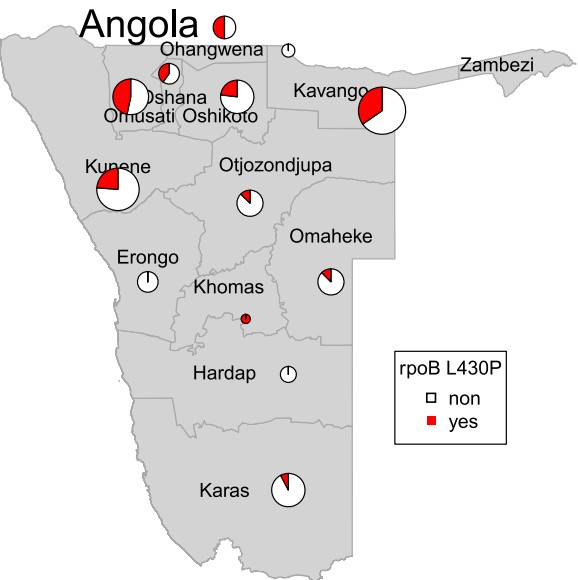

**FIG 3** Map of Namibian districts with the appearance of the mutation *rpoB* L430P. The mutations appear in all districts; however, the frequency is higher in the northern parts of the region.

All 25 isolates of the largest cluster, cluster 22 (cl22), belonged to LAM sublineage 4.3.2 and were at least MDR, thus representing 20% of all MDR strains in the study population. Three cl22 isolates were already classified as pre-XDR. Cluster 22 isolates were characterized by several shared resistance mutations, e.g., *katG* S315T, *pncA* L35P, and *rpsL* K43R, confirming their close relationship and a common ancestor. However, considering mutations in *rpoB* and *embB*, different mutations occurred (e.g., *rpoB* L430P [*n* = 22] and *rpoB* S450W [*n* = 2]), pointing toward the existence of different cl22 subbranches. To visualize these, we performed a high-resolution phylogeny and cluster analysis based on a distance threshold of 5 SNPs (d5) (Fig. 2).

Within cl22, 3 subbranches with 10 (d5 group 18), 4 (d5 group 20), and 2 (d5 group 2) isolates, respectively, could be identified (Fig. 2). The cl22 substructure identified correlates with the mutation patterns of the isolates in the subbranches. For example, the isolates of the largest d5 cluster, d5 group 18 (*n* = 10), showed a unique mutation profile, with all strains harboring a combination of the *rpoB* mutations L430P and F424L and a mutation in *rv0678* (Rv0678 D88G), a gene that is known to be the key role-player in bedaquiline resistance (12). Among the strains analyzed, 14 had mutations in *rv0678*, so more than 70% of these mutations appeared exclusively in this d5 cluster. The details are presented below.

As outlined above, 22 of the 25 cl22 isolates carried the borderline RIF resistance mutation *rpoB* L430P. Likewise, there were two other clusters (cl9 [*n* = 5] and cl2 [*n* = 3]) where strains were also characterized by this particular *rpoB* mutation. Overall, *rpoB* L430P appeared mainly in clustered isolates (31 out of 35; 88.6%), pointing toward the high transmission potential of strains with this mutation. Spatial analysis confirmed that Mtbc strains with *rpoB* L430P were found in all regions of Namibia, although the proportion of strains with this mutation was increased in the northern part of the country (Fig. 3).

The second largest d12 cluster, cl20, consisted of 10 isolates, which belonged to LAM sublineage 4.3.4.1. The cl20 isolates were characterized by the mutation combination *fabG1* −15c>t, *katG* S315T (INH resistance mutation), *rpoB* S450L (RIFr mutation), and *embB* M306V (EMB resistance mutation).

**Other mutations in resistance- and fitness-associated genes.** To further understand resistance development and possible epistatic interactions between different mutations, we determined all mutations emerging in resistance or compensatory mutation-associated genes in addition to the canonical high-confidence resistance mutations (Table S2).

When INH was considered, it became obvious that of 43 strains with the *fabG1* −15c>t mutation, 36 acquired a further mutation, of which *inhA* I21T (*n* = 21) and *katG* S315T (*n* = 12) were the most frequent. These data indicate that strains with the borderline resistance mutation *fabG1* −15c>t are frequently acquiring a secondary mutation potentially leading to a higher level of resistance, while this is not the case for strains with the high-level resistance mutation *katG* S315T, as this mutation occurred in 77 strains as the only INH resistance-related mutation (Table S2).

For RIF resistance, the situation appeared to be similar: while strains with borderline RIFr mutations such as *rpoB* L430P, H445D/L/N/Y, or D435Y acquired a wide diversity of secondary mutations in *rpoB*, this was not the case for strains with the high-level RIFr mutation S450L (Table 2; Table S2). When the 33 strains with the most frequent borderline mutation, *rpoB* L430P, were considered, the most common secondary mutations were *rpoB* F424L (*n* = 10) and H445Q (*n* = 13). While the first mutation is not listed in the recent WHO *Catalogue of Mutations in Mycobacterium tuberculosis Complex and Their Association with Drug Resistance* (13), the latter is listed as a "group 2 mutation," which is classified with uncertain significance, and "associated with resistance—interim" due to expert rule. The fact that the H445Q mutation emerged independently in two different branches in our phylogeny (d12 SNP cl9 [*n* = 5] and cl22 [*n* = 7], with one ungrouped) (Table S2) points toward a potential functional consequence. Still, eight of the nine strains with the *rpoB* L430P/H445Q combination and available pDST data were phenotypically RIF susceptible (Table 2; Table S2). This appeared to also be the case for strains with the *rpoB* L430P/F424L combination, as eight out of nine strains with pDST data available tested susceptible (Table 2; Table S2). Accordingly, both mutations may be considered compensatory rather than resistance mutations.

We screened for mutations in *rv0678* potentially associated with BDQ resistance. In total, 14 strains had mutations in *rv0678*, namely, D88G (*n* = 13) and Q131_del (*n* = 1). Interestingly, Rv0678 D88G occurred in strains of two clusters, d12 cl22 (*n* = 10) and cl23 (*n* = 3), which belonged to two different Mtbc lineages (LAM and X type, respectively). The three cl23 strains were classified as pre-XDR and had the mutation *rrs* 1401a>g rendering them resistant to injectable drugs. In combination with a functional BDQ resistance mutation, the strains would be XDR according to the new WHO classification with full first-line drug, FQ, and BDQ resistance. Unfortunately, at the time of the study, no phenotypic BDQ resistance testing was possible in Namibia.

In addition to resistance mutations, 57 strains (41.9%) had a mutation in *rpoA* or *rpoC* known to have a compensatory effect. While mutations in *rpoA* and *rpoC* were observed in Mtbc strains from several clusters, isolates from particular clusters often carried the same compensatory mutation (Table S2). For example, the *rpoC* L507V mutation (*n* = 8) was present in all strains from d12 cl18 and in three out of four strains from d12 cl1. The most frequent compensatory mutation, *rpoC* I491S, appeared 10 times in our population and was specifically present in all isolates of d12 cl20. Compensatory mutations in *rpoC* appear nearly exclusively in combination with the high-confidence mutation *rpoB* S450L (46 out of 49), two with *rpoB* S450W and one with *rpoB* V170F. We observed a similar result for compensatory mutations in *rpoA*, with 5 out of 8 connected to *rpoB* S450L, 1 connected to *rpoB* S450W, and 2 connected to *rpoB* D435V. Overall, 74% of all strains with the *rpoB* S450L mutation had an additional compensatory mutation in either *rpoA* or *rpoC*. No compensatory mutations in the upstream region of *ahpC*, compensating for fitness costs of *katG* mutations, were found.

## DISCUSSION

We present the first study employing WGS for an in-depth investigation of clinical RIFr Mtbc strains from Namibia. The results obtained demonstrate high rates of resistance to

**TABLE 2** RIF resistance mutations and secondary mutations in *rpoB* in combination with pDST results for RIF

| RIFr mutation(s) | No. of strains[a] |
|---|---|
| *rpoB* S450L | **69** |
| Not done | **10** |
| *rpoB* S450L | 9 |
| *rpoB* S450L; *rpoB* P45S | 1 |
| Resistant | **56** |
| *rpoB* S450L | 49 |
| *rpoB* D103D; *rpoB* S450L | 1 |
| *rpoB* P45L; *rpoB* S450L | 2 |
| *rpoB* S450L; *rpoB* P45S | 1 |
| *rpoB* S450L; *rpoB* A1075A | 2 |
| *rpoB* S450L; *rpoB* S576L | 1 |
| Sensitive | **3** |
| *rpoB* S450L | 3 |
| | |
| *rpoB* S450W | **5** |
| Resistant | **5** |
| *rpoB* S450W | 5 |
| | |
| *rpoB* S441L | **1** |
| Not done | **1** |
| *rpoB* S441L | 1 |
| | |
| *rpoB* V170F | **1** |
| Resistant | **1** |
| *rpoB* V170F | 1 |
| | |
| *rpoB* 437_del_aac | **1** |
| Resistant | **1** |
| *rpoB* 1309_del_aac | 1 |
| | |
| *rpoB* H445Y | **8** |
| Resistant | **8** |
| *rpoB* H445Y | 5 |
| *rpoB* H445Y; *rpoB* A1075A | 2 |
| *rpoB* H445Y; *rpoB* T427P | 1 |
| | |
| *rpoB* H445D | **1** |
| Resistant | **1** |
| *rpoB* H445D | 1 |
| | |
| *rpoB* H445L | **1** |
| Sensitive | **1** |
| *rpoB* H445L | 1 |
| | |
| *rpoB* H445N | **5** |
| Not done | **2** |
| *rpoB* H445N | 1 |
| *rpoB* H445N; *rpoB* T215N; *rpoB* S431G | 1 |
| Resistant | **1** |
| *rpoB* H445N | 1 |
| Sensitive | **2** |
| *rpoB* H445N | 1 |
| *rpoB* H445N; *rpoB* V113I; *rpoB* G456S | 1 |
| | |
| *rpoB* D435F | **3** |
| Not done | **1** |
| *rpoB* D435F; *rpoB* D875G | 1 |
| Resistant | **1** |
| *rpoB* D435F | 1 |
| Sensitive | **1** |
| *rpoB* D435F | 1 |

**TABLE 2** (Continued)

| RIFr mutation(s) | No. of strains[a] |
|---|---|
| *rpoB* D435V | **3** |
| Resistant | **3** |
| *rpoB* D435V | 1 |
| *rpoB* D435V; *rpoB* Q432E | 2 |
| | |
| *rpoB* D435V; *rpoB* H445Y | **1** |
| Resistant | **1** |
| *rpoB* D435V; *rpoB* H445Y; *rpoB* V170L | 1 |
| | |
| *rpoB* D435Y | **2** |
| Sensitive | **2** |
| *rpoB* D435Y | 2 |
| | |
| *rpoB* L430P | **33** |
| Not done | **7** |
| *rpoB* L430P | 2 |
| *rpoB* L430P; *rpoB* F424L | 1 |
| *rpoB* L430P; *rpoB* H445Q | 4 |
| Resistant | **6** |
| *rpoB* L430P | 1 |
| *rpoB* L430P; *rpoB* F424L | 1 |
| *rpoB* L430P; *rpoB* H445Q | 1 |
| *rpoB* L430P; *rpoB* I491L | 2 |
| *rpoB* L430P; *rpoB* I491M | 1 |
| Sensitive | **20** |
| *rpoB* L430P | 3 |
| *rpoB* L430P; *rpoB* F424L | 7 |
| *rpoB* L430P; *rpoB* E721K; *rpoB* R816R | 1 |
| *rpoB* L430P; *rpoB* H445Q | 8 |
| *rpoB* L430P; *rpoB* T122A; *rpoB* F424L | 1 |
| | |
| *rpoB* L430P; *rpoB* D435G | **1** |
| Resistant | **1** |
| *rpoB* L430P; *rpoB* D435G | 1 |
| | |
| *rpoB* L430P; *rpoB* D435Y | **1** |
| Resistant | **1** |
| *rpoB* L430P; *rpoB* D435Y; *rpoB* D634G | 1 |

[a]Boldface are the numbers for the overarching categories.

first-line drugs, with roughly 50% of the strains investigated being resistant to all first-line drugs and 13% of the MDR Mtbc strains being classified as pre-XDR. This is linked to high cluster rates as an indication of the ongoing recent transmission of RIFr, MDR, and pre-XDR Mtbc strains. We also observed a remarkable diversity of resistance mutations against the first-line drugs INH and RIF, with a high proportion of strains having so-called borderline resistance mutations, e.g., *inhA* promoter mutations or *rpoB* L430P. This led to the misclassification of 25% of the RIFr strains as being susceptible by pDST. Finally, we determined a potentially new BDQ resistance mutation in *rv0678*.

While our sample size is comparably small, it allows several relevant observations important for DR TB control in the country and interesting insights into the emergence and spread of resistance in relation to borderline resistance mutations not present in other areas of the world such as Eastern Europe (9, 14–17).

In our convenience sample of RIFr Mtbc strains obtained from 136 patients from 2016 to 2018 living in 13 out of 14 regions of Namibia, we demonstrated high rates of resistance to first-line drugs, in line with emerging pre-XDR and the ongoing recent transmission of MDR Mtbc clones. Indeed, 30% of all MDR strains (*n* = 35) could be found in the two largest clusters, cl20 and cl22, and 75% of all MDR strains were grouped into clusters. Looking at pre-XDR strains exclusively, 85% appeared in clusters. This points to the so far unrecognized transmission of MDR Mtbc strains as a major driver of the MDR TB epidemic in the country and argues strongly for the rapid

implementation of interventions addressing MDR TB transmission. Delayed case finding, delayed or insufficient diagnostics, and incomplete treatment are the likely reasons for ongoing MDR TB transmission in Namibia and other areas of the world (18–21). As such, immediate efforts are needed to strengthen local case detection, diagnostics, and MDR TB treatment capacities by introducing rapid comprehensive molecular resistance diagnostics based on targeted genome sequencing that could be performed on sputum samples and are able to detect resistance to WHO group A MDR TB drugs (22). This appears to be particularly important in light of the high rates of resistance to other first-line drugs among MDR Mtbc strains, the already high rate of pre-XDR strains, and the emergence of BDQ resistance due to strains with mutations in *rv0678*.

Still, rates of resistance to EMB and PZA, recommended for short-course MDR TB treatment (23), are variable in individual MDR Mtbc strains in our collection, strongly arguing for rapid early comprehensive diagnostics to allow the design of the most effective treatments. This is particularly important for patients with pre-XDR TB, for whom treatment options are rather limited, as exemplified by the already "totally resistant" strains of cl23, if the Rv0678 mutation is functional, leading to additional BDQ resistance.

One particular finding leading to a discrepancy between genotypic and phenotypic DST results is the high prevalence of Mtbc strains with borderline RIF resistance mutations that were resistant according to molecular assays such as Xpert MTB/Rif but tested mainly susceptible by pDST in the standard Bactec MGIT 960 system. This has caused uncertainty in treatment decisions, which should be overcome by clear guidance for the interpretation of molecular data, now enabled by the first catalogue of resistance-associated mutations by the WHO (13) and by a change of the critical concentration from 1.0 mg/L for RIF to 0.5 mg/L, which was recently proposed (24).

On the other hand, the high prevalence of Mtbc strains with borderline resistance mutations also opens the possibility of overcoming resistance with high-dose treatment (25, 26). This option has been suggested for several drugs such as INH, RIF, and BDQ, as several mutations vary in the levels of resistance that they confer, and may indeed be an option for patients with difficult-to-treat MDR TB with advanced resistance patterns (25, 26). However, it should be taken into account that the success of these strains, selected in Africa in particular, is most likely related to host genetic factors that lead to faster drug degradation and higher bioavailability (27–29). It will therefore be necessary to combine pathogen and host genetics/phenotypes to explore the potential of high-drug dosing on an individual-patient basis.

The link between borderline INH and RIF resistance and additional mutations points toward either resistance-enhancing or compensatory functions. Indeed, although the *rpoB* H445Q mutation has already been reported to be potentially resistance conferring (13), we could not confirm this effect in our study, as the majority of strains with the *rpoB* L430P/H445Q combination still tested susceptible. The same was observed for strains with the *rpoB* L430P/F424L combination. The *rpoB* F424L mutation has been reported in a limited number of strains globally and is not included in the WHO mutation catalogue. Accordingly, both mutations may rather have compensatory fitness-enhancing functions.

Resistance to new and repurposed drugs such as BDQ, delamanid, or linezolid threatens recent advances in the treatment of MDR/XDR TB. Mutations in *rv0678* are frequently associated with an increased MIC for BDQ and cross-resistance to clofazimine (12, 30, 31). The detection of the D88G mutation in *rv0678* is of major concern, as it was selected in two completely independent outbreak strains in our cohort. While we were not able to perform comparative BDQ MIC testing for strains with and without the D88G mutation, the independent acquisition of this mutation points toward positive selection and a functional consequence. Furthermore, we recently identified a frameshift mutation at the same codon position by selection experiments (32). Considering the use of BDQ as a first-line drug for MDR TB since 2017, the immediate implementation of advanced molecular assays and pDST is essential in Namibia.

In conclusion, our findings clearly demonstrate that urgent additional measures are necessary to control the DR TB epidemic in Namibia. To stop the ongoing transmission and further evolution of MDR/pre-XDR strains in the country, the implementation of rapid case detection, timely resistance testing for new and repurposed drugs, and genomic surveillance is essential. Currently, pDST for new and repurposed drugs is not performed in Namibia. Universal pDST for patients with DR TB should become standard and linked with optimized individualized treatment concepts. The development of an implementation strategy for individualized treatment regimens should be a programmatic priority. The combination of pathogen and host genetics for drug dosing regimens in cases of borderline resistance-conferring mutations appears to be a valid approach, opening new pathways for the treatment of patients with MDR/pre-XDR TB. However, such an approach needs to be tested in prospective clinical trials and is resource-intensive. Resource allocation and effective collaborations between local and international stakeholders need to be further strengthened to achieve the programmatic goals of TB control. The concepts developed in Namibia could be used as a model for other countries with a similar structure. To achieve this, we are undertaking a prospective surveillance study on RIFr TB to identify hot spots of transmission, and we have implemented targeted genome sequencing for rapid genomic DST in the country. Studies integrating pathogen and host pharmacogenomics to test tailor-made drug regimens are planned.

## MATERIALS AND METHODS

**Study design.** A convenience sample of 136 Mtbc strains from individuals with at least RIFr TB diagnosed by Xpert MTB/RIf (Cepheid, Sunnyvale, CA) from 2016 to 2018 was collected for WGS analysis from all available strains at the Namibia Institute of Pathology Limited (NIP) strain repository. All RIFr Mtbc strains in the country are collected at the NIP in Windhoek, the only laboratory where culture and drug susceptibility testing (DST) are performed in the country.

**Data collection.** Routine clinical data related to the patients whose strains had been selected were collected from eTB Manager, an electronic, web-based data collection tool for drug-resistant TB of the National TB and Leprosy Programme (NTLP).

**Strain characterization.** Culture and pDST for first- and second-line drugs were performed at NIP using the Bactec MGIT 960 system according to the manufacturer's instructions (Becton, Dickinson and Company, Franklin Lakes, NJ). Chromosomal DNA was extracted using the cetyltrimethylammonium bromide (CTAB) method as described previously (33).

**Whole-genome sequencing.** WGS was performed on an Illumina NextSeq 500 machine using the Nextera XT library preparation kit according to the manufacturer's instructions (Illumina, San Diego, CA). Raw read data (fastq files) were mapped to the *M. tuberculosis* H37Rv reference genome (GenBank accession number NC_000962.3) using BWA-MEM (34), and mapping was refined with the GATK software package (35). All data sets had a minimum mean genome-wide coverage of the H37Rv reference genome of at least 50-fold. Variants (single nucleotide polymorphisms [SNPs] and insertions and deletions [indels]) were detected based on the SAMtools pileup file (34), and variants with a minimum coverage of 4 reads in both the forward and reverse orientations, 4 reads calling the allele with a Phred score of at least 20, and an allele frequency of 75% were considered confident variant calls. Multiple consecutive SNP calls (in a 12-bp window), which could reflect artificial variant calls around indels, drug resistance-associated genes, and repetitive regions (e.g., PPE/PGRS genes), were excluded.

All remaining SNPs that had a clear base call for all strains and matched the above-mentioned threshold levels in at least 95% of all strains were considered valid and used for a concatenated sequence alignment.

**WGS data analysis.** A maximum likelihood (ML) tree was constructed using IQ-TREE software (36) with the ModelFinder option and ascertainment bias correction. The consensus tree was rooted with the midpoint root option in FigTree v1.4.4 (37), and nodes were arranged in increasing order. Phylogenetic Mtbc lineages were inferred from specific SNPs based on data reported previously by Coll et al. (7).

Strains were grouped into "SNP clusters" by considering maximum pairwise genetic distances between at least two Mtbc isolates of ≤5 SNPs and ≤12 SNPs as an indicator of TB cases associated with direct transmission events, as suggested previously by Roetzer et al. and Walker et al. (38, 39).

Polymorphisms for 27 drug resistance-associated genes that are involved in mechanisms of drug resistance to any of the drugs included in the MDR/XDR TB regimen (see Tables S1 and S2 in the supplemental material) and 3 compensatory target genes (*rpoA* and *rpoC*, suggested to compensate for *in vitro* growth deficits of RIFr strains mediated by *rpoB* mutations [40], and the *ahpC* upstream region, suggested to compensate for a catalase [*katG*] deficit in isoniazid-resistant strains [41]) were determined from the WGS data using a previously published interpretation catalogue (Table S1) (42).

Isolates with known resistance markers (Table S1) were considered resistant to the respective antibiotics.

**Statistics.** Descriptive statistics were calculated (means, SD, and proportions) for all variables as applicable.

**Ethics approval.** The study received approval from the Ministry of Health and Social Services Research Office in Namibia and the University of Namibia Research Ethics Committee, Windhoek, Namibia (H-G Campus/450/2019).

**Data availability.** The sequencing data analyzed for this study have been deposited in the European Nucleotide Archive (ENA) database (accession number PRJEB54067).

## SUPPLEMENTAL MATERIAL

Supplemental material is available online only.

**SUPPLEMENTAL FILE 1**, XLSX file, 0.1 MB.
**SUPPLEMENTAL FILE 2**, XLSX file, 0.1 MB.
**SUPPLEMENTAL FILE 3**, PDF file, 0.1 MB.

## ACKNOWLEDGMENTS

We thank all of the patients who participated in this study. We also thank the laboratory staff at the Molecular and Experimental Mycobacteriology Laboratory, Borstel, Germany, and the Namibian Institute of Pathology Limited Windhoek, Namibia, for their great support of the laboratory work. In particular, we are grateful to Tanja Niemann, Tanja Struve-Sonnenschein, and Vanessa Mohr for providing excellent technical assistance.

We declare no conflict of interest.

The study was supported by the SeqMDRTB_NET project funded by the German Ministry of Health in the framework of the Global Health Protection program. Additional support was provided by the Deutsche Forschungsgemeinschaft (DFG) (German Research Foundation) under Germany's Excellence Strategy for Excellenz Cluster Precision Medicine in Chronic Inflammation EXC 2167, by the German Ministry of Education and Research (BMBF) for the German Center of Infection Research (DZIF), and by the Leibniz Science Campus of Evolutionary Medicine of the Lung 431 (EvoLUNG).

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
