## [Reviewer comments · Microbiology Spectrum]

Microbiology Spectrum

Whole genome sequencing for resistance prediction and transmission analysis of *Mycobacterium tuberculosis* complex strains from Namibia

Mareli Claassens, Emmanuel Nepolo, Viola Dreyer, Queendeline Mokomele, Gerhard Van Rooyen, Nunurai Ruswa, Gunar Günther, and Stefan Niemann

Corresponding Author(s): Stefan Niemann, Forschungszentrum Borstel

Review Timeline:

Submission Date:	May 5, 2022
Editorial Decision:	June 22, 2022
Revision Received:	August 15, 2022
Accepted:	August 29, 2022

Editor: Luiz Pedro de Carvalho

Reviewer(s): The reviewers have opted to remain anonymous.

Transaction Report:

DOI: <https://doi.org/10.1128/spectrum.01586-22>

June 22, 2022

Dr. Stefan Niemann
Forschungszentrum Borstel
Molecular Mycobacteriology
Parkallee 1
23845 Borstel
Germany

Re: Spectrum01586-22 (Whole genome sequencing for resistance prediction and transmission analysis of Mycobacterium tuberculosis complex strains from Namibia)

Dear Dr. Stefan Niemann:

Link Not Available

Sincerely,

Luiz Pedro Carvalho

Journals Department
Reviewer comments:

Reviewer #1 (Public repository details (Required)):

It is not obvious that this dataset has been made publicly available but I believe it should be, as in other studies of its kind

Reviewer #1 (Comments for the Author):

Very well written, clear, paper on the extremely high TB burden in Namibia, a novel and useful contribution to the TB community.

Minor amendments and suggestions below:

Suggest adding some stats re the association of the main mutations (such as L430P) with lineage, for instance. Even though the numbers are low, it looks to be significant

Wondering why the Authors used d12 rather than d5 in such a high burden setting for the overall phylogeny, would imagine d5 more appropriate

Line 31 - 'the strains investigated were (rather than being) resistant'

Line 97 - mapping (rather than mappings)

Line 144 - Figure 1 (rather than Figure 2?)

Line 202 - (d5 group, n=18) rather than (d5 group 18) etc.

Line 286 - suggest 'incomplete treatment' rather than 'improper treatment'

Line 296-299 - was pDST performed for BDQ in this case? It would be good to have that info

Line 301 - 'borderline RIF resistance mutations' rather than 'low-level RIF resistance mutations' - this is the language that is being used in the literature for these mutations so suggest sticking with that

Line 318-320 was pDST performed with CC of 1.0 or 0.5ug/ml here, should specify

Lines 499 and 506 - equal to twelve SNPs, rather than equal than twelve SNPs

Supplementary tables should be included (as supplementary) as they have lots of useful information for the reader. They could be labelled and explained a little better.

Line 295 - 'arguing for rapid', rather than 'arguing rapid'

Reviewer #2 (Public repository details (Required)):

The authors need to deposit the sequencing data and provide of the project number in the main text

Reviewer #2 (Comments for the Author):

The present manuscript analyze a strain set of 136 M. tuberculosis isolates from Namibia. Understanding the genomics diversity in high burden countries can help to local TB control efforts. This is a well conducted analysis demonstrating high clustering rate and high MDR resistance. Particularly interesting is the finding of a high diversity of RIF resistance conferring mutations compared to other countries, mainly in Europe, where MDR is rampant but dominated by few clones with similar genetic make up. The results are relevant but I have some comments which I hope can help the authors.

1. While relevant it is still not clear how representative is the sample set. On one hand around Namibia has around 12000 new cases/year. Here we are looking to 136 strains from 2016-2018. Therefore it is not clear the representativeness. In that sense if will be good to know how likely is that the sample is bias towards drug resistance or whether the number of clustered cases is artefactually increased by selection biases. The authors mention a convenience sample but even being convenient some explanation is needed to understand representativeness of the results

2. One interesting finding is a high diversity of known RIFr mutations as well as the identification of others mutations which are borderline and/or poorly characterized. This fact suggests failures in the treatment follow up programme. At the same time I wonder if there is a link between those mutations and HIV status (which is know to alter drug concentrations dyanmics)? Something in the lines of what has been described here: <https://pubmed.ncbi.nlm.nih.gov/32718966/>

3. It is not clear to me if DST for all strains is available and how it was carried out (assumed as part of the NTP). Please specify as it is not mentioned in methodology but then it is mentioned in several parts of the manuscript to corroborate genotyping data

4. Bedaquiline resistance. Can the authors explain whether this particular patient was treated with BDQ or at least the use (or not) of BDQ in the country? What I am trying to understand is whether any of these three scenarios is possible: 1. Naïve patient with mutations that may cause a higher MIC to BDQ; 2. Naïve patients which acquired transmitted BDQ strain or 3. Patient who acquired BDQ during treatment

5. Compensatory rpoC mutations -> can the authors test if they are more likely associated to clustering than any other mutations randomly selected. Also, are they associated to S450L or do you see compensatory mutations associated to some of the rare RIF conferring mutations

6. Figure 1. Please use the same notation for lineages/sublineages than in the main text. Also better to remove branch lengths as they are not legible
7. Please deposit and provide the sequencing project number in the main text

Staff Comments:

Preparing Revision Guidelines

Please return the manuscript within 60 days; if you cannot complete the modification within this time period, please contact me. If you do not wish to modify the manuscript and prefer to submit it to another journal, please notify me of your decision immediately so that the manuscript may be formally withdrawn from consideration by Microbiology Spectrum.

Reviewer #1 (Public repository details (Required)):

It is not obvious that this dataset has been made publicly available but I believe it should be, as in other studies of its kind

Thank you for this comment and of course the data needs to be available. We have uploaded the sequencing data to the European Nucleotide Archive under the study accession PRJEB54067. We also added a chapter "Data availability" to the manuscript.

Reviewer #1 (Comments for the Author):

Very well written, clear, paper on the extremely high TB burden in Namibia, a novel and useful contribution to the TB community.

We thank the reviewer for this very supportive comment.

Minor amendments and suggestions below:

Suggest adding some stats re the association of the main mutations (such as L430P) with lineage, for instance. Even though the numbers are low, it looks to be significant

We agree that a correlation of main mutations e.g. with lineage would be an interesting finding. However, although there is a significant association between rpoB L430P and lineage 4, we didn't want stress this observation as this effect might be biased due to the unequal distribution of lineages in collection spanning a limited timeframe only.

Wondering why the Authors used d12 rather than d5 in such a high burden setting for the overall phylogeny, would imagine d5 more appropriate

Indeed, we worked with both thresholds. We used the d12 distance to get a first insight into longer more ancient transmission networks that reflect the longitudinal transmission of particular clones/strains and then zoomed into these larger groups by subdividing them with d5 distance. This approach was suggested in previous publications e.g. Roetzer et. al. Genome Medicine 2013 and Walker et al. Lancet Infectious Diseases 2013.

Line 31 - 'the strains investigated were (rather than being) resistant'

changed

Line 97 - mapping (rather than mappings)

changed

Line 144 - Figure 1 (rather than Figure 2?)

Correct, sorry for the mistake. Changed it to figure 1.

Line 202 - (d5 group, n=18) rather than (d5 group 18) etc.

Here, the reviewer misunderstood our writing. We named our d5 groups with numbers. We changed it to "group_18" to make this fact clearer for the readers. We also changed the legend of Figure 2, accordingly.

Line 286 - suggest 'incomplete treatment' rather than 'improper treatment'

changed

Line 296-299 - was pDST performed for BDQ in this case? It would be good to have that info

Phenotypic DST to BDQ was not performed in those cases. There is no capacity until now in Namibia to do pDST for BDQ and other novel drugs. In selected cases samples are send to South Africa.

Line 301 - 'borderline RIF resistance mutations' rather than 'low-level RIF resistance mutations' - this is the language that is being used in the literature for these mutations so suggest sticking with that

We agree with the reviewer and changed the term throughout the manuscript.

Line 318-320 was pDST performed with CC of 1.0 or 0.5ug/ml here, should specify

The DST was done with 1.0 mcg/ml.

Lines 499 and 506 - equal to twelve SNPs, rather than equal than twelve SNPs

changed

Supplementary tables should be included (as supplementary) as they have lots of useful information for the reader. They could be labelled and explained a little better.

We will include them and agree that they have lots of useful information, e.g. d5 clustering included.

Line 295 - 'arguing for rapid', rather than 'arguing rapid'

changed

Reviewer #2 (Public repository details (Required)):

The authors need to deposit the sequencing data and provide of the project number in the main text

Thank you for this comment and of course the data needs to be available. We have uploaded the sequencing data to the European Nucleotide Archive under the study accession PRJEB54067. We also added a chapter "Data availability" to the manuscript.

Reviewer #2 (Comments for the Author):

The present manuscript analyze a strain set of 136 M. tuberculosis isolates from Namibia. Understanding the genomics diversity in high burden countries can help to local TB control efforts. This is a well conducted analysis demonstrating high clustering rate and high MDR resistance. Particularly interesting is the finding of a high diversity of RIF resistance conferring mutations compared to other countries, mainly in Europe, where MDR is rampant but dominated by few clones with similar genetic make up. The results are relevant but I have some comments which I hope can

help the authors.

1. While relevant it is still not clear how representative is the sample set. On one hand around Namibia has around 12000 new cases/year. Here we are looking to 136 strains from 2016-2018. Therefore it is not clear the representativeness. In that sense it will be good to know how likely is that the sample is biased towards drug resistance or whether the number of clustered cases is artefactually increased by selection biases. The authors mention a convenience sample but even being convenient some explanation is needed to understand representativeness of the results

We are sorry, that the point was not clear in the paper. Indeed, we did not focus on all culture positive TB cases in Namibia, we only included Rif resistant cases. Therefore, the paper is on purpose focused on Rif resistant cases only as this was our intention. Still, we only capture a fraction of the RIF resistant cases in the country - Namibia recorded about 300 – 400 RIFr cases of TB before COVID 19. Last year it were just over 200. As we selected a random collection of RIFr strains from the Namibia Institute of Pathology Limited (NIP) strain repository, we believe that our strain set, acknowledging the risk of selection bias, is representative. We also agree with the reviewer that prospective population based data are essential to constantly monitor the drug resistance development in the country. Therefore, we initiated a prospective molecular surveillance of all available DR M. tuberculosis complex cases in 2021.

2. One interesting finding is a high diversity of known RIFr mutations as well as the identification of others mutations which are borderline and/or poorly characterized. This fact suggests failures in the treatment follow up programme. At the same time I wonder if there is a link between those mutations and HIV status (which is known to alter drug concentrations dynamics)? Something in the lines of what has been described here: <https://pubmed.ncbi.nlm.nih.gov/32718966/>

Indeed, the diversity of mutations esp. the presence of “low level” resistance mutations is an interesting finding that may be related to host factors and co-infections such as HIV. In our study there was no association of HIV status and the specific low-level RIF mutations.

3. It is not clear to me if DST for all strains is available and how it was carried out (assumed as part of the NTP). Please specify as it is not mentioned in methodology but then it is mentioned in several parts of the manuscript to corroborate genotyping data

According to the Namibian diagnostic algorithm from 2017, phenotypic DST is done for Rifampicin, Isoniazid, Moxifloxacin, Levofloxacin, Ethambutol and Amikacin in cases, which show resistance to Rifampicin in the GenXpert MTB/RIF. Genotypic DST is done using Genotype MTBDRsl. Samples not demonstrating resistance to Rifampicin don't undergo routinely DST. Unfortunately, stock outs interrupt this work flow regularly.

4. Bedaquiline resistance. Can the authors explain whether this particular patient was treated with BDQ or at least the use (or not) of BDQ in the country? What I am trying to understand is whether any of these three scenarios is possible: 1. Naïve patient with mutations that may cause a higher MIC to BDQ; 2. Naïve patients which acquired transmitted BDQ strain or 3. Patient who acquired BDQ during treatment

Bedaquiline was first introduced in TB treatment in Namibia in 2017. The cases reported with D88G mutation started treatment between 10/2016 and 02/2018. None had FQ resistance. 9/14 were new TB cases. This all makes a previous exposure to BDQ in the cases reported very unlikely. All this

makes it most likely, that D88G is a mutation in naïve patients, which might be associated with higher MIC. Our plan is to research this further, while building local capacity for MIC testing. We added the sentence to the discussion: “Unfortunately at the time of the study no phenotypic BDQ resistance testing was possible in Namibia.”

5. Compensatory *rpoC* mutations -> can the authors test if they are more likely associated to clustering than any other mutations randomly selected. Also, are they associated to S450L or do you see compensatory mutations associated to some of the rare RIF conferring mutations

We thank the reviewer for this comment, and agree that these questions are of importance. Indeed, more than 80% of the strains with compensatory mutations are clustered (d12; 54% d5) and compensatory mutations in *rpoC* appear nearly exclusively with *rpoB* S450L (46/49), besides that we only have 2 with *rpoB* S450W and one with *rpoB* V170F. We added this fact to the manuscript: “Compensatory mutations in *rpoC* appear nearly exclusively in combination with the high confidence mutation *rpoB* S450L (46 out of 49), two with *rpoB* S450W and one with *rpoB* V170F. We observed a similar fact with compensatory mutations in *rpoA*, with five out of 8 connected to *rpoB* S450L, one to *rpoB* S450W and two with *rpoB* D435V. Overall, 74% of all strains with the *rpoB* S450L mutation have an additional compensatory mutation in either *rpoA* or *rpoC*.”

6. Figure 1. Please use the same notation for lineages/sublineages than in the main text. Also better to remove branch lengths as they are not legible

We agree that the subgroups would be handy, but we intentionally left them out for better visibility, otherwise we would have to use too many groups and thus colours. Removed the branch lengths.

7. Please deposit and provide the sequencing project number in the main text

done

Staff Comments:

Preparing Revision Guidelines

- Point-by-point responses to the issues raised by the reviewers in a file named "Response to Reviewers," NOT IN YOUR COVER LETTER.
- Upload a compare copy of the manuscript (without figures) as a "Marked-Up Manuscript" file.
- Each figure must be uploaded as a separate file, and any multipanel figures must be assembled into

one file.

- Manuscript: A .DOC version of the revised manuscript
- Figures: Editable, high-resolution, individual figure files are required at revision, TIFF or EPS files are preferred

August 29, 2022

Dr. Stefan Niemann
Forschungszentrum Borstel
Molecular Mycobacteriology
Parkallee 1
23845 Borstel
Germany

Re: Spectrum01586-22R1 (Whole genome sequencing for resistance prediction and transmission analysis of Mycobacterium tuberculosis complex strains from Namibia)

Dear Dr. Stefan Niemann:

Your manuscript has been accepted, and I am forwarding it to the ASM Journals Department for publication. You will be notified when your proofs are ready to be viewed.

Sincerely,

Luiz Pedro Carvalho
Editor, Microbiology Spectrum

Journals Department
Supplemental Material: Accept
Supplemental Table S2: Accept
Supplemental Table S1: Accept